# An Edge Based Multi-Agent Auto Communication Method for Traffic Light Control

**DOI:** 10.3390/s20154291

**Published:** 2020-07-31

**Authors:** Qiang Wu, Jianqing Wu, Jun Shen, Binbin Yong, Qingguo Zhou

**Affiliations:** 1School of Information & Engineering, Lanzhou University, Lanzhou 730000, China; wuq17@lzu.edu.cn (Q.W.); yongbb@lzu.edu.cn (B.Y.); 2School of Computing and Information Technology, University of Wollongong, Wollongong 2522, Australia; jw937@uowmail.edu.au (J.W.); jshen@uow.edu.au (J.S.)

**Keywords:** ITS, IoT, reinforcement learning, MRAL, multi-agent, MAAC, edge computing

## Abstract

With smart city infrastructures growing, the Internet of Things (IoT) has been widely used in the intelligent transportation systems (ITS). The traditional adaptive traffic signal control method based on reinforcement learning (RL) has expanded from one intersection to multiple intersections. In this paper, we propose a multi-agent auto communication (MAAC) algorithm, which is an innovative adaptive global traffic light control method based on multi-agent reinforcement learning (MARL) and an auto communication protocol in edge computing architecture. The MAAC algorithm combines multi-agent auto communication protocol with MARL, allowing an agent to communicate the learned strategies with others for achieving global optimization in traffic signal control. In addition, we present a practicable edge computing architecture for industrial deployment on IoT, considering the limitations of the capabilities of network transmission bandwidth. We demonstrate that our algorithm outperforms other methods over 17% in experiments in a real traffic simulation environment.

## 1. Introduction

Traffic congestion has caused a series of severe negative impacts like longer waiting time, more gas cost, and severe air pollution. According to a report in 2014 [1], the loss caused by traffic jams is up to $124 billion US dollars a year in the US. The shortage of traffic infrastructures, the growing number of vehicles, and the inefficient traffic signal control are key underlying reasons for traffic congestion. Among these, the traffic light control problem seems to be the most easily solved. However, the internal operation of the real urban transportation environment cannot be accurately calculated and analyzed mathematically due to its complexity and uncertainty. Reinforcement learning (RL), which is characterized by being data-driven, mode-less, and self-learning, is well suited for conducting research on adaptive traffic light control algorithms [2,3,4].

The rapid development of artificial intelligence technology and deep learning (DL) has played a vital role in many fields. In recent years, DL has gained great success in image classification [5,6,7,8], machine translation [9,10,11,12], healthcare [13], smart city [14], time-series forecast [15], Game of Go [16] etc. The intelligent transportation systems (ITS) also have benefited from the latest AI achievement.

Traditional adaptive traffic light control method [2,3] could achieve local optimization by adapting to single intersection based on RL. Furthermore, global optimization is needed to achieve dynamic multi-intersection control in large smart city infrastructure. Multi-agent reinforcement learning (MARL) is increasingly being used to study more complex traffic light control issues [17,18,19].

Although the existing methods have effectively improved the control efficiency of traffic signal control, they still have the following problems: (1) shortage of communication between a traffic light and other traffic lights; (2) shortage of consideration of the limitations of the capabilities of network transmission bandwidth. The contributions of this paper are summarized as the following:We present an auto communication protocol (ACP) between agents in MARL based on attention mechanism;We propose a multi-agent auto communication (MAAC) algorithm based on MARL and ACP in traffic light control;We build a practicable edge computing architecture for industrial deployment on Internet of Things (IoT), considering the limitations of the capabilities of network transmission bandwidth;The experiments show the MAAC framework outperformed 17 %over baseline models.

The remainder of this paper is organized as follows: Section 2 introduces related works including multi-agent system, RL, IoT, edge computing, and the basic concept of communication theory. Section 3 formulates the definition of the traffic light control problem. Section 4 details the MAAC model and our edge computing architecture for IoT. Section 5 conducts the experiments in a traffic simulation environment and demonstrates the results of the experiments with a comparison between our methods and others. Section 6 concludes the paper and discusses future work.

## 2. Related Work

Urban traffic signal control theory has been continuously investigated and developed for nearly 70 years since the 1950s. However, from theory to practice, the goal of alleviating urban traffic congestion through the optimization and control of urban traffic signals, is consisting in very complex control problems. Urban traffic signal control is to allocate the time of a signal cycle and the ratio of the time of red and green lights in a signal cycle. The control methods include fixed time [20], vehicle detection [21], and automatic control [22]. The fixed time and vehicle detection methods cannot adapt to the dynamic changes in traffic flow and complex road conditions. The automatic control methods are difficult to implement for its high algorithm complexity.

A multi-agent system is an important branch of distributed AI research, with the ability of distribution, autonomy, coordination, learning, and reasoning [23]. In 1989, Durfee et al. [24] proposed the use of a negotiation mechanism to share tasks among multiple agents. In 2007, Marvin Minsky argued that human thoughts were constructed by multi-agents [25]. In 2016, Sukhbaatar et al. [18] and Hoshen et al. [19] observed all agents using a centrally controlled method in local environment, and then output the probability distribution of multi-agents’ joint actions. Alibaba and University College London (UCL) proposed the use of two-way communication network between agents network (BiNet) [26], and achieved good results in the StarCraft game mission in 2017. From the trend of the multi-agent system, communication between agents has gotten more and more attention.

Adaptive traffic light control [2,3] is a relatively easy way to ease the traffic jam for smart city. Although adaptive traffic light control methods have achieved local optimization by adapting to single intersection based on RL (single-agent), the city has thousands of traffic lights. Thus, the global traffic light optimization could be considered as a multi-agent system, which has been studied by Chen et al. [27]. Moreover, the deployment structure must be taken into consideration for the industrial deployment.

Here is the summary of the methods of traffic signal control (as Table 1 shown):

### 2.1. Reinforcement Learning

#### 2.1.1. Single-Agent Reinforcement Learning

Single-agent reinforcement learning was developed to train one agent, which chose a series of actions to get more rewards after interacting with an environment. To learn an optimal policy for the agent to gain maximal reward is the aim of the algorithm. At each time step *t*, the agent interacts with the environment to maximize the total reward $R_{T}$, where *T* is the total number of time steps of an episode until it finishes. The rewards obtained after each action being performed are accumulated, which would be: $R_{T}=r_{1}+r_{2}+\ldots +r_{T}$.

RL algorithm, which has the characteristics of “data-driven, self-learning, and model-free”, is considered to be a practical method to solve problem of traffic light control [28,29]. As shown in Figure 1.

The traffic signal is regarded as an “agent” with decision-making ability at the intersection. By observing the real-time traffic flow, the current traffic status $S_{t}$ and the reward $R_{t}$ is obtained. According to the current status, the agent selects and executes the corresponding action (change lights or keep). Then, the agent observes the effect of the action on the intersection traffic to obtain the new traffic state $S_{t+1}$ and the new reward $R_{t+1}$. The agent evaluates the action just selected so that it executes, optimizes strategies until converging to the optimal “state and action”.

#### 2.1.2. Multi-Agent Reinforcement Learning

All agents apply their actions to the environment for whole rewards. From this perspective, we define a multi-agent reinforcement learning (MARL) environment as a tuple $\left(X_{1}-A_{1},X_{2}-A_{2}\ldots ,X_{m}-A_{m}\right)$ where $X_{m}$ is any given agent and $A_{m}$ is any given action, then the new state of the environment is the result of a set of joined actions defined by $A_{1},A_{2},\ldots ,A_{n}$. In other words, the complexity of MARL scenarios increases with the number of agents in the environment (as shown in Figure 2).

The urban traffic signal control can be seen as a typical multi-agent system. With the traditional adaptive traffic signal control method based on RL, the new signal controls have been expanded from one intersection to multiple intersections.

MARL is mainly to study the cooperative and coordinated control actions of multiple states of intersections, which extend the single-agent RL algorithm to multi-agents in the urban traffic environment. The MARL based methods in this field are divided into three categories [30]: (1) a completely independent MARL at each intersection; (2) MARL in cooperation with some states from the intersections; (3) MARL in all states from the intersections.

The collaboration mechanism is an important part of MARL in traffic signal control at multiple intersections. Each agent could estimate the action probability model of other agents without real-time computing, but it was still difficult to update the estimation model in a dynamic environment [31].

### 2.2. Attention Mechanism

Attention mechanism has recently been widely used in various fields of deep learning (for example, image processing [32], natural language processing [12,33].), achieving good results. From a conceptual perspective, attention imitates human cognitive methods, selectively filters out a small amount of important information, and focuses on this important information, ignoring most unimportant information. The attention information selection process is reflected in the calculation of information weight coefficients.

The specific calculation method is divided into three steps (as shown in Figure 3):(1)Calculate the similarity or correlation between Query and Key;
(1)$$\mathrm{Similarity}(\mathrm{Query},\;\left.{\mathrm{Key}}_{i}\right)=\mathrm{Query}\cdot {\mathrm{Key}}_{i}$$(2)Normalize the result calculated in step (1) to obtain the weighting coefficient;
(2)$$a_{i}=Softmax\left(Similarity_{i}\right)$$(3)The weighting coefficient is used to perform weighted sum on Value.
(3)$$\mathrm{Attention}(\mathrm{Query},\;\mathrm{Source})\;=\sum_{i=1}^{L_{x}}a_{i}\cdot {\mathrm{Value}}_{i}$$

### 2.3. IoT

With the development of Wireless Sensor Network (WSN) [34] and 5th-Generation (5G) communication technologies [35], the IoT connects millions of devices, including vehicles, smartphones, home appliance, and other electronics, enabling these objects exchange data [36]. The traditional Internet has been extensively up-scaled via IoT [37,38,39]. As shown in Figure 4.

#### Cloud and Edge Computing

Cloud computing is a paradigm for enabling ubiquitous, convenient, on-demand network access to a shared pool of configurable computing resources (e.g., networks, servers, storage, applications, and services), which can be rapidly provisioned and released with minimal management efforts or service provider interaction.

With cloud computing, edge computing emerges as a novel form of computing paradigm, which shifts from centralized to decentralized [40]. Compared with conventional cloud computing, it provides shorter response times and better reliability. To save bandwidth and reduce the latency, more data is processed at the edge rather than uploaded to the cloud. Thus, mobile devices of users can complete parts of the workload at the edge of the network. Similarly, in modern transportation, edge devices can be deployed on roadsides and in vehicles for better communications and control between connected objects.

## 3. Preliminary

### 3.1. Multi-Agent Communication Model

The multi-agent communication model (as shown in Figure 5) is in accordance with Shannon communication model [41], the applied perception and behavior of agents can be modeled by information reception and transmission. The agent acts as a communication transceiver, and the internal structure information of the agent is encoded and decoded. The environment is the communication channel between the agents. In actual modeling, a continuous matrix is generally used for multi-agent communication [18,42].

#### 3.1.1. Shannon Communication Model

The basic problem of communication is to reproduce a message sent from one point to another point. In 1948, Shannon proposed the Shannon communication model [41], which represented the beginning of modern communication theory. Shannon communication model is a linear communication model, consisting of six parts: sender, encoder, channel, noise, decoder, and receiver, as shown in Figure 6:

#### 3.1.2. Communications Protocol

The communication protocol [43] is also called the transmission protocol. Both parties involved in the communication carried out end-to-end information transmission according to the agreed rules, and both parties can understand the received information. The communication protocol is mainly composed of grammar, semantics, and timing. The syntax includes the data format, encoding, and signal level; the semantics represent the data content that contains control information, and the timing represents clear rate matching and sequencing of communications.

### 3.2. Problem Definition

In the problem of multi-agent traffic signal control, we consider it as a Markov Decision Process (MDP): <x,π,R,γ>, where x the state of all intersections; π is the policy to create actions; R is reward from all crossroads; γ is the discount factor. Furthermore, we define: each agent that controls the change (duration) of traffic lights is $Agent^{i}\left(i\in N\right)$; $\pi_{i}$ is $Agent^{i}$ for all acceptable traffic light duration control strategies, rewarding the $R_{i}$ environment for the level of traffic congestion at the intersection of $Agent^{i}$ and other Agents ($Agent^{-i}$ and its policy $\pi_{-i}$) (It can be calculated according to the specific indicators of vehicle queue length, the lower the congestion level, the greater the reward), $\left(c_{1},\ldots ,c_{N}\right)$ is the communication matrix *C* between agents, so in our multi-agent traffic signal problem, the objective functions controlled are:(4)$$R_{i}\left(x;\pi_{i},\pi_{-i},C\right)=E\left[\sum_{t=0}^{-\infty }-\gamma_{i}^{t}r_{i}\left(x_{t},\pi_{i,t},\pi_{-i,t},C_{t}\right)\right]$$

In the Eqution (Equation 4), $\pi_{i}$ is the policy of $Agent^{i}$; $\pi_{-i}$ is the policy of $Agent^{-i}$; *t* is the timestep. The problem is to find a better strategy to maximize the value of the above formula.

## 4. Methodology

In this section, we will detail the multi-agent auto communication (MAAC) model and an edge computing architecture for IoT.

### 4.1. MAAC Model

In the MAAC model (as shown in Figure 7):

Each agent can be modeled by distributed, partially observable Markov decision (Dec-POMDP). The strategy of each agent ($\pi_{\theta_{t}}$) is generated by a neural network. In each time step, the agent will observe the local environment $x_{t}$ and the communication information sent by other agents $\left(c^{1},\ldots ,c^{i-1},c^{i+1},\ldots ,c^{N}\right)$. Through the combination of the above time series information, the Agent generates the next action $(a_{t+1}$ and the next communication message $c_{t+1}$ sent out by the internal processing mechanism (parameter is $\theta_{i}$).

The joint actions of all Agents ($a_{1},\ldots ,a_{N}$ ) interact with the environment, which is to obtain the maximum value of the centralized value function (θ=E[R]). The MAAC algorithm is designed to improve the neural network parameter set $\theta_{i}$ of each agent through the process of optimizing the central value function. The overall architecture of the MAAC model can be regarded as a distributed MARL model with automatic communication capabilities.

#### 4.1.1. Internal Communication Module in Agent

The internal communication module (ICM) in an agent is an important part in MAAC model (as shown in Figure 8).

Each Agent, which is divided into two sub-modules, with the receiving end and the sending end. The receiving end receives the information of other agents and uses the attention mechanism for information processing, and then sends the processed information to the sending end; the sending end observes the external environment and uses the information processed by the receiving attention mechanism to generate information using a neural network.

Receiving End$Agent^{i}$ will use the attention mechanism to filter information received from other Agents ($Agent^{-i}$). Firstly it generates its own message $c^{t}$ from a combined message $C=c^{1},\ldots ,c^{N}$ after receiving the information of $Agent^{-i}$. Then, it picks important messages and ignores unimportant ones. Herein, we introduce the parameter set $W^{q}$,$W^{k},W^{v}$, which are calculated separately (could be calculated in parallel):
(5)$$\begin{array}{cc} \; & q^{i}=W^{q}\cdot C\\ \; & k^{i}=W^{k}\cdot C\\ \; & v^{i}=W^{v}\cdot C \end{array}$$Then, we calculate the information weight $\hat{\alpha^{i}}=softmax\left(q^{i}{\dot{k}}^{i}\right)$. Finally we get the weighted information after the information selection: $\hat{C}=\sum_{i=1}^{N}\alpha_{i}c_{i}$. Sending EndThe sending end of the Agent receives the information of other Agents processed by the Attention mechanism of the receiving end $\hat{C}$, and through the observed local environment $x_{t}$, generates the next execution action through the neural network $a_{t+1}$ and communication information $c_{t+1}$.

#### 4.1.2. MAAC Algorithm

At the time of *t* in the MAAC model, the environment input is $X_{t}=\left(x_{t}^{1},\ldots ,x_{t}^{N}\right)$ and corresponding communication information input is $C_{t}=\left(c_{t}^{1},\ldots ,c_{t}^{N}\right)$. Multi-agents ($Agent_{1},\ldots ,Agent_{N}$) are going to interact with each other. Each Agent receives information with receivers and transmitters internally. The receiver receives its own environmental information $x_{t}$ and communication information $c_{t}$, and generates action and external interaction information group $\left(a_{t+1},c_{t+1}\right)$ at t+1. The MAAC model collects all agent actions to form a joint action ($a_{1},\ldots ,a_{N}$), interacting with the environment and optimizing objective strategy for each agent.
(6)$$\nabla_{\theta_{i}}V\left(\theta_{i}\right)=E\left[\nabla_{\theta_{i}}\log\pi_{\theta_{i}}\left(a_{i}^{t}|c_{i}^{t}\right){\hat{Q}}_{t}\left(,a_{t}^{1},\ldots ,a_{t}^{N}\right)\right]$$

The calculation steps of MAAC at time *t* are shown in the Figure 9:

In the MAAC algorithm (as shown in Algorithm 1), the parameter set of $Agent^{i}$ for each agent is $\theta^{i}$. Furthermore, $\theta^{i}$ is divided into the sender $\theta_{Sender}^{i}$ and receiver $\theta_{Receiver}^{i}$. The parameters of the sending end and the receiving end, which are optimized by the overall multi-agent objective function, iteratively updating the parameter set of the receiver and the sender in the communication module of each agent.
**Algorithm 1** MAAC learning algorithm process1: Initialize the communication matrix of all agents $C_{0}$2: Initialize the parameters of the agent $\theta_{Sender}^{i}$ and $\theta_{Receiver}^{i}$3: **repeat**4:  Receiver of $Agent^{i}$: uses attention mechanism to generate communication matrix $\hat{C_{t}}$5:  Sender of $Agent^{i}$: chooses an action $a_{t+1}^{i}$ from policy selection network, or randomly chooses   action a (e.g.,ϵ-greedy exploration)6:  Sender of $Agent^{i}$: generates its own information through the receiver’s communication matrix   $\hat{C_{t}}$$c_{t+1}^{i}$7:  Collect all the joint actions of Agent and execute the actions $a_{t+1}^{1},\ldots ,a_{t+1}^{N}$, get the reward from   the environment $R_{t+1}$ and next state $X_{t+1}$8:  Update the strategic value function of each Agent:   $\nabla_{\theta_{i}}V\left(\theta_{i}\right)=E\left[\nabla_{\theta_{i}}\log\pi_{\theta_{i}}\left(a_{i}^{t}|c_{i}^{t}\right){\hat{Q}}_{t}\left(,a_{t}^{1},\ldots ,a_{t}^{N}\right)\right]$9: **until** End of Round Episode10: returns $\theta_{Sender}^{i}$ and $\theta_{Receiver}^{i}$ for each Agent

### 4.2. Edge Computing Structure

In order to deploy MAAC algorithms in an industrial scale environment, we must take the network delay into consideration. We propose an edge computing architecture near every traffic light. An edge computing device needs to have the following functions: (1) it could detect vehicles’ information (location, direction, velocity) from the surveillance video of its intersection in real-time and record the vehicle information; (2) it could run the traffic signal control algorithm to control the traffic light nearby (see Figure 10).

## 5. Experiments

In this section, we first built the urban traffic simulator based on our edge computing architecture. Then, we have applied the MAAC algorithm and other baseline algorithms to the simulation environment for comparing the performance of all models.

### 5.1. Simulation Environment and Settings

We apply an open source simulator for traffic environment: CityFlow [44] as our experiment environment. We assumed that there are six traffic lights (intersection nodes or edge computing nodes) in one section of a city (as shown in Figure 11).

Our dynamic control of the traffic lights was using the CityFlow [44] Python interface at runtime.

Here are the settings of our experiments (as shown in Table 2).

The directionsOne traffic light at $node_{0}$ has four neighbor nodes $\left(node_{1},node_{2},node_{3},node_{4}\right)$, four entries (in), and four exits (out). The road length is set to 350 m and vehicle speed limit is set to be 30 (km/h).Traffic light agentWe apply traffic signal control algorithm into a docker container [45].Communication delay settingThe communication delay from center to a traffic light is set as 1 s (sleep 1 s in the code).Traffic control timing cycleWe initially set a traffic light time cycle as 45 s, and green light interval $g_{t}=20$ s, red light interval $r_{t}=20$ s, and yellow light interval $y_{t}=5$ s.EpisodeOne episode time is set as 15 min (900 s), including 20 traffic light time cycles.Vehicle simulation settingWe assume vehicles arrive at road entrances according to the Bernoulli process with the random probability $P_{in}=\frac{1}{15}$ at one intersection. Every vehicle has a random destination node except for the entry node (we set random(seed)=7). In one episode, there are approximately 400 vehicles.Hyper-parameter settingThe learning rate is set to 0.001; γ is set to 0.992; the reward is the average waiting time at intersection.

### 5.2. Baseline Methods

Fix-timeIn this method we set all the traffic light timing as fixed traffic control timing cycle as we have mentioned in the experiments.Q-learning (Center)Q-learning algorithm [46] is deployed on center (docker) to generate traffic light control action. The delay from the traffic light agent to an intersection is set at 1.0 s.Q-learning (Edge)Q-learning algorithm [46] herein is deployed on edge device (docker) to generate traffic light control action. The delay from the traffic light agent to an intersection is set 0.1 s.Nash Q-learningNash Q-learning [47] extends Q-learning to a non-cooperative MARL. An agent maintains Q-functions over joint actions, and performs updates based on assuming Nash equilibrium behavior over the current Q-values.

### 5.3. Evaluation

The time of a vehicle enters an entry of the intersection until it passes through, is defined as $t_{M}$, where *M* is the number of vehicles. In simulations, we record the time for all vehicles at one intersection in every episode. At last, we accumulate all the times record over all the intersections, $T_{e}=\sum_{i=1}^{I}\sum_{m=1}^{M}t_{m}$. To evaluate the traffic network, where *E* is the number of the episode, *M* is the number of vehicles, and *I* is the number of intersections that every vehicle will pass through.

### 5.4. Results

We have applied five methods, including Fixed-time method [20], Q-learning (Center) [46], Q-learning (Edge) [46], Nash Q-learning [47], and our MAAC method. They were all trained in 1000 episodes in CityFLow [44] based on edge computing architecture as we designed. As shown in Figure 12, we can see that the the algorithms converged at around 600 episode point. The MAAC method performed the fastest convergence in the training process, comparing with other models.

After trainning, we tested the algorithms in 500 episodes after the training process. As shown in Table 3, the MAAC performed the best among the traffic signal control algorithms.

As shown in Table 4, our method did not sacrifice the waiting time of some intersections to ensure overall performance. Furthermore, the performance of every intersection was optimized at different levels. From the method Q-learning (center) and Q-learning (edge), we can see that the edge computing structure has reduced the network delay for the deployment environment.

As shown in Table 5, the delay time and delay rate of Q-learning (Center) are the highest, which proves the edge computing structure we proposed is useful for reducing the network delay. The MAAC still outperforms others when computing the delay time of the network.

## 6. Conclusions

In this work, we proposed a multi-agent auto communication (MAAC) algorithm based on the multi-agent reinforcement learning (MARL) and an auto communication protocol (ACP) between agents with the attention mechanism. We built a practicable edge computing structure for industrial deployment on IoT, considering the limitations of the capabilities of network transmission bandwidth.

In the simulation environment, the experiments have shown the MAAC framework outperformed 17% over baseline models. Moreover, the edge computing structure is useful for reducing the network delay when deploying the algorithm on an industrial scale.

In future research, we will build a simulation environment much closer to the real world and take the communication from vehicle to traffic light into consideration to improve the MAAC method.

## Figures and Tables

**Figure 1 sensors-20-04291-f001:**
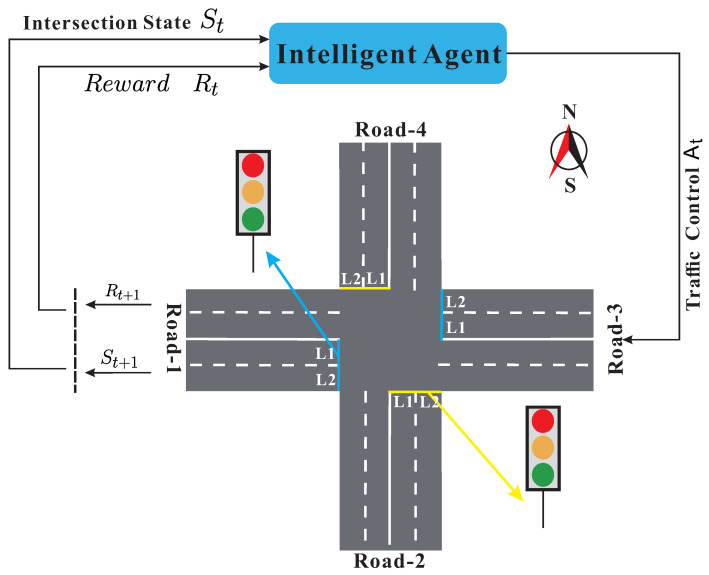
An illustration of an intersection road.

**Figure 2 sensors-20-04291-f002:**
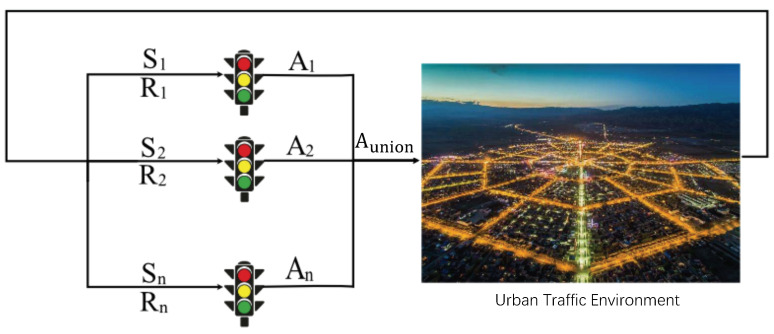
The multi-agent reinforcement learning (MARL) structure in urban traffic signal control.

**Figure 3 sensors-20-04291-f003:**
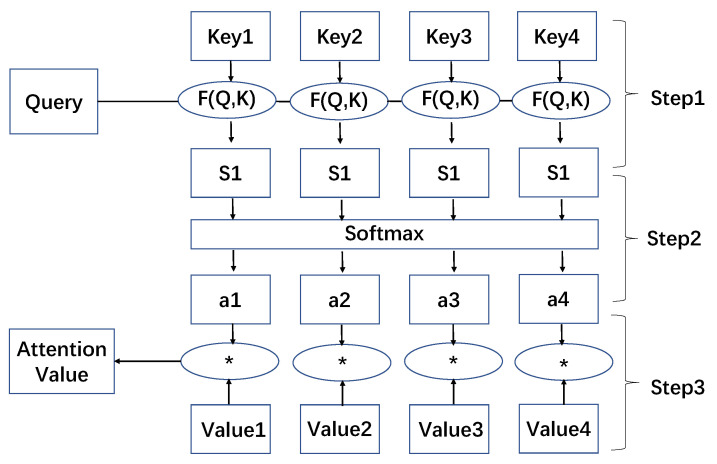
The method of computing attention.

**Figure 4 sensors-20-04291-f004:**
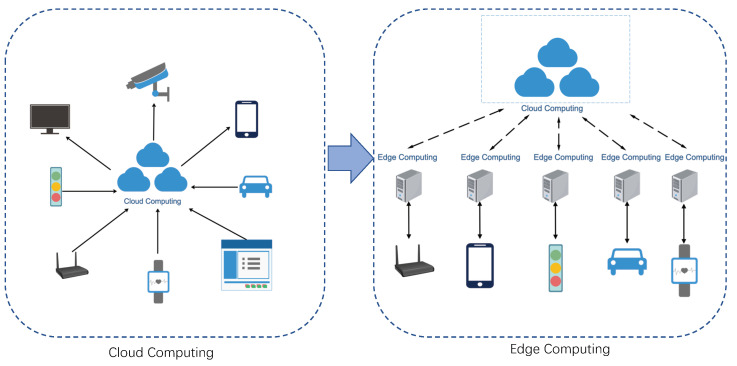
The structure of cloud computing and edge computing.

**Figure 5 sensors-20-04291-f005:**
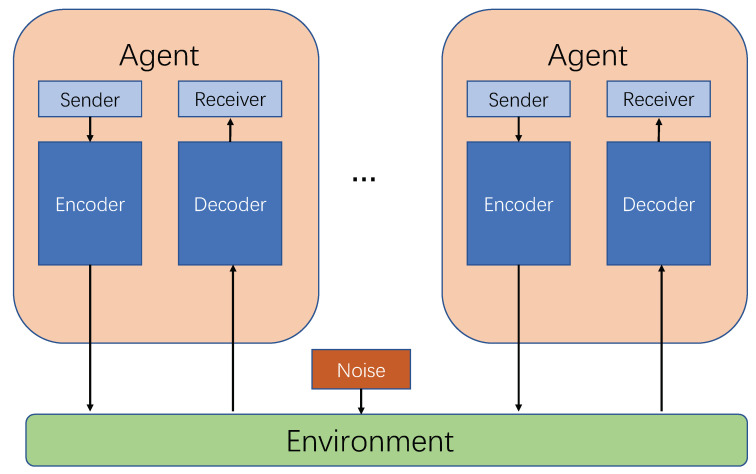
The structure of multi-agent communication model based on Shannon communication model.

**Figure 6 sensors-20-04291-f006:**
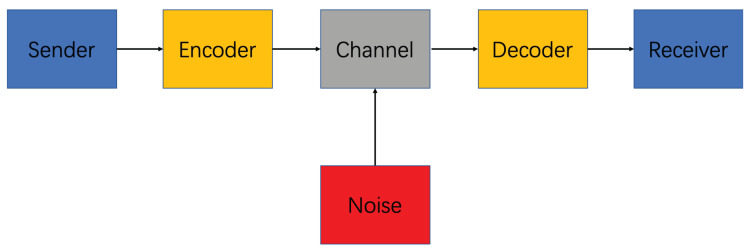
An illustration of Shannon communication model.

**Figure 7 sensors-20-04291-f007:**
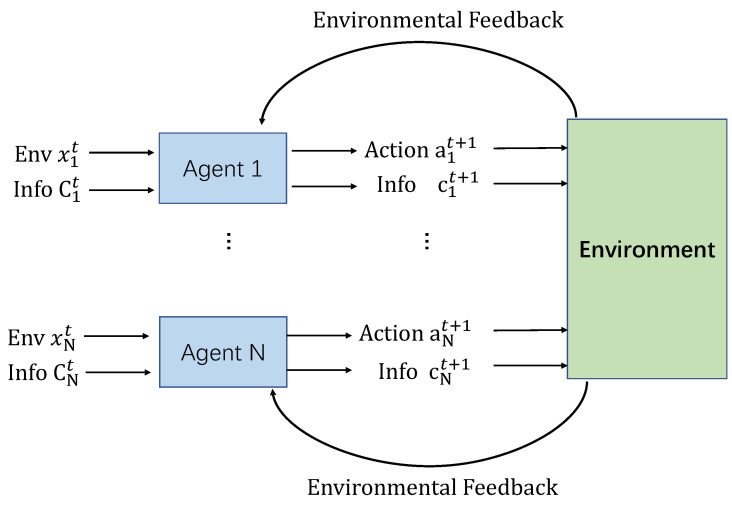
The structure of multi-agent auto communication (MAAC) model.

**Figure 8 sensors-20-04291-f008:**
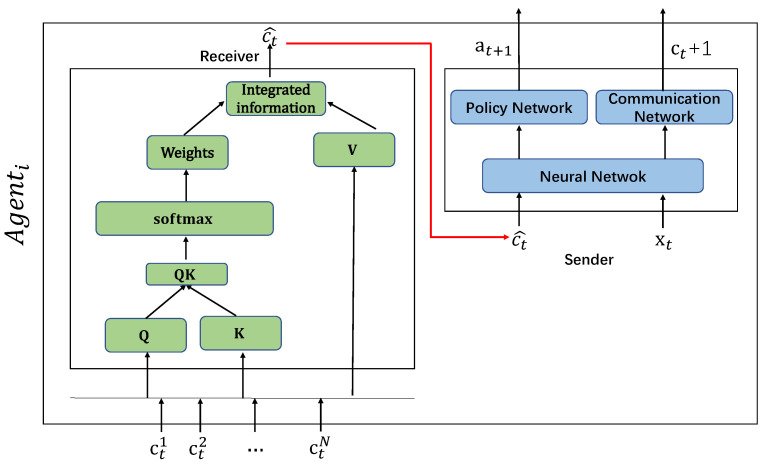
The internal communication module in an agent.

**Figure 9 sensors-20-04291-f009:**
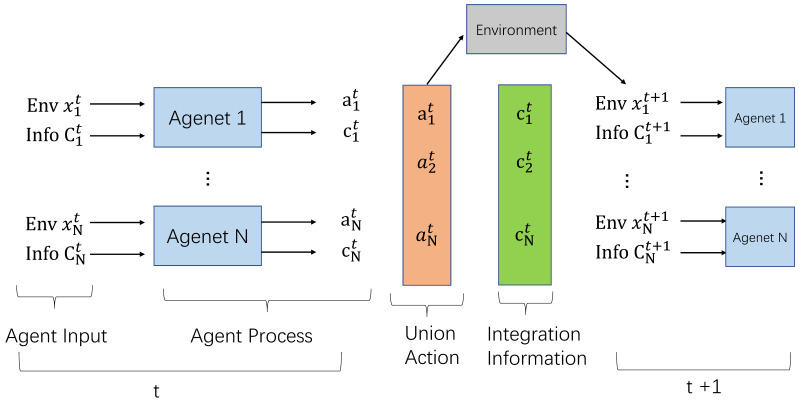
The process of MAAC algorithm compute.

**Figure 10 sensors-20-04291-f010:**
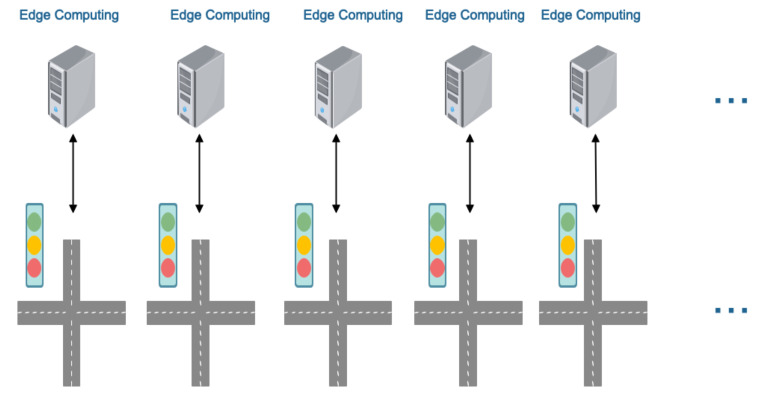
The edge devices are deployed near the traffic lights.

**Figure 11 sensors-20-04291-f011:**
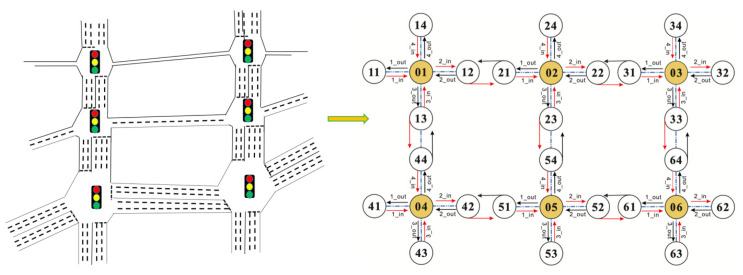
The experiment environment for multi-intersection traffic signal control.

**Figure 12 sensors-20-04291-f012:**
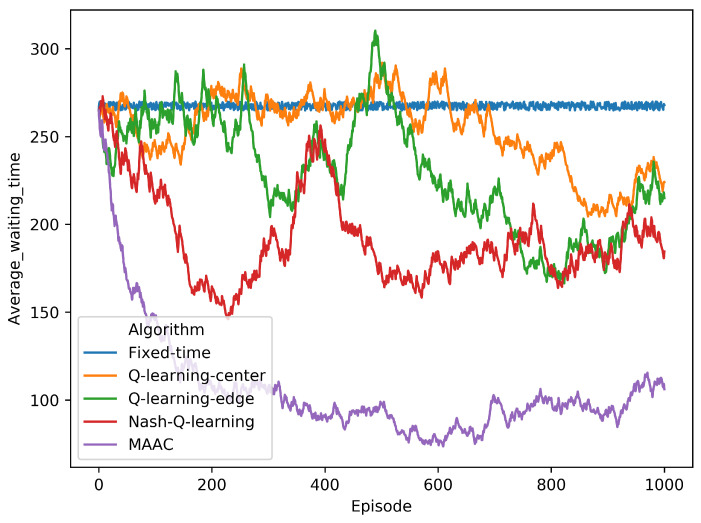
The training process of five methods.

**Table 1 sensors-20-04291-t001:** The summary of the methods of traffic signal control.

Method	Pros	Cons
Fixed time [20]	Easy to deploy and implement, still the mainstream method today.	Inability to dynamically adapt to intersection changes.
Optimize one traffic light [2,3]	The traffic signal can be adjusted according to the dynamic changes of the intersection situation.	Urban traffic signals are actually composed of multiple intersections, and the local optimization of a single intersection cannot represent the overall optimization of multiple intersections.
Optimize multiple traffic lights [27]	Global optimization of traffic signals at multiple intersections in a city.	It is difficult to implement and deploy, and the algorithm also has room for optimization, such as considering multi-agent communication.

**Table 2 sensors-20-04291-t002:** The detailed simulation parameters.

Simulation Parameters	Value
Road length	350 (m)
Vehicle speed limit	30 (km/h)
Traffic control timing cycle	$g_{t}=20$, $r_{t}=20$, $y_{t}=5$ (s)
Episode	900 (s)
Vehicle simulation setting	400 (one episode)

**Table 3 sensors-20-04291-t003:** The resluts of five methods.

Method	The Average Velocity (km/h)	The Average Waiting Time (s)
Fixed-time	11.15	166.71
Q-learning (center)	17.44	135.64
Q-learning (edge)	19.11	112.24
Nash-Q-learning	23.12	90.75
MAAC	26.42	80.21

**Table 4 sensors-20-04291-t004:** The experiment results of average waiting time at each intersection.

	Result	Fixed-Time	Q-Learning (Center)	Q-Learning (Edge)	Nash-Q-Learning	MAAC
ID						
1		156.37	146.69	126.74	103.00	76.09
2		188.65	199.72	135.21	108.54	83.90
3		155.29	136.11	123.34	104.54	74.15
4		197.33	178.63	111.52	102.50	79.49
5		155.57	160.85	123.11	94.52	84.12
6		168.45	139.90	132.77	100.33	62.33

**Table 5 sensors-20-04291-t005:** The delay time of five methods.

Method	Average Episode(s)	Average Delay(s)	Delay Rate
Fixed-time	38,827.5	0.0	0.0%
Q-learning (Edge)	35,789.3	7522.9	21.1%
Q-learning (Center)	45,448.0	20,809.7	45.8%
Nash Q-learning	31,340.4	5951.6	18.9%
MAAC	28,940.5	3551.6	12.2%

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
