# Peer review of "An Edge Based Multi-Agent Auto Communication Method for Traffic Light Control"

_sensors, 2020, doi:10.3390/s20154291_

Round 1

Reviewer 1 Report

This paper presents a multi-agent reinforcement learning based traffic light control method considering auto communication features. It is an interesting and practical topic. Few comments are as follows,

  1. The related work section needs to be revised. The section involved too many hot topics which are irrelevant regarding this work mainly focused on traffic light control strategy. Especially, the IoT subsection and the Cloud and Edge Computing subsection are too general and non-academic.
  2. Certain definitions for notations are not clearly stated. For example, in Equation 4, PIi and PI-i are introduced, but what are the definition for notation t, PIi,t and PI-i,t? The authors need to carefully revise the notations and their definitions.
  3.  For section 4.2, the authors pointed out an edge computing structure, but such structure is not relevant too any aforementioned information. Again, it seems irrelevant to this study. If the authors think this part is an academic contribution, they need to give reasons. 
  4. In terms of the result analysis, since this paper considered the communication delay in the experiment, the authors may need to analyze the impact of the delay quantitatively. Current evaluation section and results analysis are too naive.

Minor issues,

In the abstract, line 11, it should be 17% rather than %17. And a few other minor grammar issues need to be revised. 

Author Response

Dear editor,

We are very grateful for your valuable comments!  We have carefully revised the paper as you suggested and checked the content of the manuscript.

Here is the response to review report 1 as follows:

1. The related work section needs to be revised. The section involved too many hot topics which are irrelevant regarding this work mainly focused on traffic light control strategy. Especially, the IoT subsection and the Cloud and Edge Computing subsection are too general and non-academic.

R: We thank the reviewer for pointing these out. We have added the papers of the traffic light control strategy in lines(68-73). Besides, the traffic light control could consider as a multi-agent system, where we have made clear this point, and the cloud and edge computing is the environment of the traffic control algorithm deployment. We provided generic introduction for readers who might feel unfamiliar as far as ‘related work’ is concerned but we actually discussed in-depth about technical aspects relevant to this work in latter parts of the paper.

2. Certain definitions for notations are not clearly stated. For example, in Equation 4, PIi and PI-i are introduced, but what are the definition for notation t, PIi,t and PI-i,t? The authors need to carefully revise the notations and their definitions.

R: Thanks for this suggestion. We have added certain definitions for notations for PIi , PI-i, and t in section 3.2. We also checked other parts thoroughly.

3. For section 4.2, the authors pointed out an edge computing structure, but such structure is not relevant too any aforementioned information. Again, it seems irrelevant to this study. If the authors think this part is an academic contribution, they need to give reasons.

R: Section 4.2 proposes an edge computing structure, which is needed for the reasons follows:

(1) The edge computing structure is the environment to deploy MAAC algorithms on an industrial scale.

(2) The edge computing structure is an IoT deployment structure for traffic light control nearby devices, which cloud reduce the network delay.

(3) In section 5.3, the results of the experiments show that the edge computing structure is useful to reduce the network delay, especially in Q-learning(center) and Q-learning(edge).

We agree edge is not the main contribution of this paper but we need to include it in our essential discussions.

4. In terms of the result analysis, since this paper considered the communication delay in the experiment, the authors may need to analyze the impact of the delay quantitatively. The current evaluation section and results analysis are too naive.Minor issues,

R: We have extended the results and analysis in Section 5. We analyzed the impact of the delay for algorithms and proved our edge computing structure played critical roles

5. In the abstract, line 11, it should be 17% rather than %17. And a few other minor grammar issues need to be revised.

R: We have revised the line 11 in the abstract, and we have revised other grammar issues.

Reviewer 2 Report

The authors propose a multi-agent auto communication algorithm that improves the adaptive global traffic light control. Is a good proposal that impact in the VANETs and Smart Cities researches.

I propose the following corrections before a final acceptation.

- Missing spaces before references (algorithms[2], smart city[4], etc.) and in "17 %over". Review the full paper.
-Improve redaction in this paragraphs "With the rapid development of artificial intelligence (AI), especially in deep learning (DL), DL has played an essential role in many fields".
-Correct the Figure.1.
-Review the comas "".
-Spaces and points in "intersections(as show in Figure.2).
-In section 3.1.2 there are a “the” repeated.
-Missing final point in Figure 8.
-Figure 7 and 9 says Agenet, should be Agent?
-Line 212 says Ane, should be One?
-Why title in Table 1 uses capital letters?, also missing final point.
-Table 2 missing final point and the title should be placed over the table.
-I recommend extend the conclusions.

Author Response

Dear editor,

We are very grateful for your valuable comments!  We have carefully revised the paper as you suggested and checked the content of the manuscript.

Here is the response to review report 2 as follows:

1. Missing spaces before references (algorithms[2], smart city[4], etc.) and in "17 %over". Review the full paper.

R: We thank the reviewer for pointing these out. We have added the spaces before references in the full paper, and also checked other typos. Sorry about this due to latex template inconsistencies.

2. Improve redaction in this paragraphs "With the rapid development of artificial intelligence (AI), especially in deep learning (DL), DL has played an essential role in many fields".

R: We have modified the sentence as "The rapid development of artificial intelligence technology and deep learning (DL) has played a vital role in many fields".

3. Correct the Figure.1.

R: We have modified the "Figure 1".

4. Review the comas "".

R: We have checked and modified comas "" in the full paper.

5. Spaces and points in "intersections(as show in Figure.2).

R: We have added spaces and deleted the points in "intersections(as shown in Figure 2)".

6. In section 3.1.2 there are a “the” repeated.

R: We have deleted the repeated "the" in 3.1.2.

7. Missing final point in Figure 8.

R: We have added the final point in Figure 8.

8. Figure 7 and 9 says Agenet, should be Agent? 

R: We have revised the Figure 7 and 9, Agent is the correct spell.

9. Line 212 says Ane, should be One?

R: We have changed the Ane to One.

10. Why title in Table 1 uses capital letters?, also missing final point.

R: We have revised the title in Table 1, and added the final point.

11. Table 2 missing final point and the title should be placed over the table.

R: We have added the final point in Table 2 (has been Table 4), and moved the title top the table.

12. I recommend extend the conclusions.

R: We have extended the conclusions, adding the function of the edge computing structure.

Reviewer 3 Report

The authors proposed a multi-agent scheme for traffic light control. Please find below my comments/suggestions.

  1. Add the comparative table of related work by highlighting the aim, proposed scheme, pros, and cons.
  2. Smart city and ITS belongs to the internet of multimedia things; therefore, I recommend to discuss briefly and add these latest related papers.

 Internet of Multimedia Things (IoMT): Opportunities, Challenges and Solutions. Sensors 202020, 2334.

  1. In Figure 5, fix the typo error of Noize.
  2. Add the MDP of the proposed scheme and explain it in the body text.
  3. Add the detailed simulation parameters in tabular form.
  4. Explain it in detail the learning rate, gamma, reward, and all other parameters used in the simulation for the proposed scheme and compared ones.
  5. When the proposed scheme converges?
  6. The results are not enough for journal paper. It needs to consider congestion, random control, longest queue, and most car models.
  7. Explain in detail the logical reasoning of the results.

Author Response

Dear editor,

We are very grateful for your valuable comments!  We have carefully revised the paper as you suggested and checked the content of the manuscript.

Here is the response to review report 3 as follows:

1. Add the comparative table of related work by highlighting the aim, proposed scheme, pros, and cons.

R: We thank the reviewer for pointing these out. We have added Table 1 to detail the pros and cons of the traffic signal control methods.

2. Smart city and ITS belongs to the internet of multimedia things; therefore, I recommend to discuss briefly and add these latest related papers.

Internet of Multimedia Things (IoMT): Opportunities, Challenges and Solutions. Sensors 2020, 20, 2334.

R: We have added the latest related papers of the internet of multimedia things in the related works([39][40]).

3. In Figure 5, fix the typo error of Noize.

R: We have fixed the typo error of "Noize" to "Noise".

4. Add the MDP of the proposed scheme and explain it in the body text.

R: We have added the MDP of the proposed scheme and explain it in section 3.2.

5. Add the detailed simulation parameters in tabular form.

R: We have added Table 2 to detail the simulation parameters.

6. Explain it in detail the learning rate, gamma, reward, and all other parameters used in the simulation for the proposed scheme and compared ones.

R: We have added hyper-parameter setting(line 242-244).

7. When the proposed scheme converges?

R: From the training results, our proposed scheme converges at around 600 episodes, we have added this clarification in the line 269-270.

8. The results are not enough for journal paper. It needs to consider congestion, random control, longest queue, and most car models.

R: We have extended this part greatly now, for example, we added Table 3, adding the average velocity, the average waiting time to illustrate the MAAC performed best.

9. Explain in detail the logical reasoning of the results.

R: Table 3 shows the MAAC performed the best among the traffic signal control algorithms; Table 4 shows our method did not sacrifice the waiting time of some intersections to ensure overall performance, the performance of every intersection was optimized at different levels; Table 5 shows the delay time and delay rate of Q-learning (Center) are the highest, which proves the edge computing structure we proposed is useful for reducing the network delay. The MAAC still outperforms others when computing the delay time of the network.

From Table 3 to Table 5, we illustrate that the MAAC algorithm improved the multiple traffic lights control. In addition, the edge computing structure is useful for reducing network delay when we take network delay into consideration. Moreover, our method is suitable for deploying on an industrial scale.

Round 2

Reviewer 1 Report

The quality of this paper has been improved. The current version is publishable. No further comments.

Reviewer 3 Report

Thank you for revising the paper according to the comments. Overall, it is greatly improved. However, the suggested  paper is not discussed. Kindly add this in the final version.

Internet of Multimedia Things (IoMT): Opportunities, Challenges and Solutions. Sensors 2020, 20, 2334.